# Hypoxia Pathways in Parkinson’s Disease: From Pathogenesis to Therapeutic Targets

**DOI:** 10.3390/ijms251910484

**Published:** 2024-09-29

**Authors:** Yuanyuan Gao, Jiarui Zhang, Tuoxian Tang, Zhenjiang Liu

**Affiliations:** 1National Engineering Laboratory for AIDS Vaccine, School of Life Sciences, Jilin University, Changchun 130012, China; yygao9921@mails.jlu.edu.cn (Y.G.); zjr24@mails.jlu.edu.cn (J.Z.); 2Department of Biology, University of Pennsylvania, Philadelphia, PA 19104, USA; tuoxian@sas.upenn.edu

**Keywords:** Parkinson’s disease, hypoxia, hypoxia pathways, neurodegenerative diseases, transmembrane protein 175, DJ-1, leucine-rich repeat kinase 2 (LRRK2)

## Abstract

The human brain is highly dependent on oxygen, utilizing approximately 20% of the body’s oxygen at rest. Oxygen deprivation to the brain can lead to loss of consciousness within seconds and death within minutes. Recent studies have identified regions of the brain with spontaneous episodic hypoxia, referred to as “hypoxic pockets”. Hypoxia can also result from impaired blood flow due to conditions such as heart disease, blood clots, stroke, or hemorrhage, as well as from reduced oxygen intake or excessive oxygen consumption caused by factors like low ambient oxygen, pulmonary diseases, infections, inflammation, and cancer. Severe hypoxia in the brain can manifest symptoms similar to Parkinson’s disease (PD), including cerebral edema, mood disturbances, and cognitive impairments. Additionally, the development of PD appears to be closely associated with hypoxia and hypoxic pathways. This review seeks to investigate the molecular interactions between hypoxia and PD, emphasizing the pathological role of hypoxic pathways in PD and exploring their potential as therapeutic targets.

## 1. Introduction

Hypoxia, or reduced oxygen supply, is a significant factor in the development of various neurodegenerative diseases, including Alzheimer’s disease (AD), Parkinson’s disease (PD), amyotrophic lateral sclerosis (ALS), and other age-related disorders [1,2,3,4,5]. The neurotoxic effects of hypoxia are linked to brain injury and the onset of neurodegenerative conditions such as AD, vascular dementia, and PD [6]. Prolonged hypoxia can lead to irreversible neuronal loss, thereby exacerbating these neurodegenerative pathologies [7].

Parkinson’s disease is primarily characterized by both motor and non-motor symptoms, along with the gradual loss of dopaminergic neurons in various brain structures, particularly the *substantia nigra pars compacta* (SNpc). Dysfunction in the SNpc results in motor symptoms such as bradykinesia (slow movements), rigidity, resting tremor, impaired postural control, and gait deficits [8,9]. Additionally, non-motor symptoms, including fatigue, cognitive impairment, sleep disturbances, anxiety, depression, and impaired autonomic regulation, significantly affect the quality of life in PD patients [10].

The diagnosis of PD is based on a thorough medical history and neurological examination, requiring the essential presence of bradykinesia alongside rigidity and/or resting tremor. These distinct motor symptoms are associated with various aspects of disease progression, complications, and disability. PD subtyping is further complicated by the evolving manifestations of the disease over time [11]. Studies have shown a lack of significant associations between diffusion metrics and most evaluated clinical symptoms, underscoring the complexity of PD pathology [12].

Non-motor manifestations, such as rapid eye movement sleep disorder, anosmia, constipation, and depression, emerge during the prodromal or premotor stage and progress alongside cognitive impairment and dysautonomia as the disease advances, often becoming predominant in the later stages [13]. Additionally, there is a connection between abnormalities in thalamostriato-hypothalamic functional connectivity and dysautonomia in PD patients [14].

Addressing these concerns, it is crucial to identify new targets or pathways for developing therapies that can alleviate symptoms or potentially combat the disease. The research has shown that hypoxia is associated with cell death mechanisms relevant to neurodegenerative diseases, particularly through the accumulation of reactive oxygen species (ROS), which are implicated in conditions such as Huntington’s disease, PD, and AD [15]. Abnormal misfolding and aggregation of alpha-synuclein in the brain are pathological features of Parkinson’s disease. Hypoxia is closely associated with the activation of alpha-synuclein, suggesting that low oxygen levels may influence the progression of PD [16]. This review article aims to explore the complex and multifaceted role of hypoxia in the pathogenesis of PD and to highlight hypoxia as a potential therapeutic target.

## 2. Hypoxia Pathways

Before discussing hypoxia pathways, it is crucial to differentiate between hypoxia training and hypoxia insult [17]. Hypoxia training involves mild, intermittent, and controlled hypoxia stimuli designed to enhance an individual’s resistance and cellular adaptation to hypoxia insult. Examples include intermittent hypoxia cycles and frequent physical exercise, which may alleviate PD symptoms by improving hypoxia pathways [16,18,19]. In contrast, hypoxia insult is typically caused by pathological processes, toxins, chronic hypoxia, or a severe decrease in ambient oxygen levels, leading to cellular injury and exacerbation of PD symptoms [1,20]. Different treatments, animal models, and in vitro models reflect various hypoxia conditions, resulting in paradoxical effects within the same pathway. Understanding the nuanced role of hypoxia can help clarify the complex evidence surrounding hypoxia pathways.

At the molecular level, low oxygen levels reduce the availability of electron acceptors in the mitochondrial electron transport chain (ETC), leading to increased ROS production and diminished ATP synthesis. These changes can directly cause neuronal damage and, in rare cases, contribute to PD symptoms [21,22,23]. Hypoxia-inducible factors (HIFs) and nuclear factor erythroid 2-related factor 2 (Nrf2) are central regulators of cellular adaptation to hypoxia, and their neuroprotective functions are well established [24,25,26]. Hypoxic exposure has been shown to induce α-synuclein(α-syn) overexpression and oligomer formation, leading to cell injury in HEK293 cells [27]. Given that hypoxia can influence the pathogenesis of PD, regulating HIF-1 and Nrf2 activity may be a crucial neuroprotective strategy for promoting the survival of dopaminergic neurons.

### 2.1. HIF Pathway

HIFs are transcription factors that respond to low oxygen levels by regulating molecular adaptations to maintain oxygen supply and energy metabolism [28]. HIFs are part of the PER-ARNT-SIM (PAS) subfamily within the basic helix–loop–helix (bHLH) family of heterodimeric transcription factors [29]. Currently, three HIF members have been identified: HIF-1, HIF-2 (also known as endothelial PAS domain-containing protein 1), and HIF-3 [30,31,32]. A Northern blot analysis reveals minimal expression of HIF-3 in brain tissue, suggesting a limited connection between HIF-3 and PD [33].

HIF-1 and HIF-2 are both heterodimeric transcription factors composed of α- and β-subunits. HIF-1α and HIF-2α share 48% amino acid sequence similarity, with 83% identity in their basic bHLH domains and approximately 70% homology in their PAS regions [34]. Additionally, the oxygen-dependent degradation domains of these HIF-α subunits, including the two critical proline residues that interact with prolyl hydroxylase (PHD), also exhibit a high degree of homology [35]. This significant sequence and structural similarity suggest that HIF-1 and HIF-2 have similar regulatory mechanisms involving PHD–von Hippel–Lindau (pVHL) complex and target genes.

Both HIF-1 and HIF-2 share the same β-subunit, HIF-1β, which is stably expressed and located in the nucleus, binding to hypoxia response elements (HREs), upstream of hypoxia-inducible genes [36,37]. Under normoxic conditions, HIF-α subunits are continuously degraded through hydroxylation, which is catalyzed by PHD and factor inhibiting HIF (FIH). Prolyl hydroxylation promotes interaction with the pVHL E3 ubiquitin ligase complex, leading to degradation by the 26S proteasome [38]. During moderate hypoxia, PHD activity is impaired, but FIH can still perform hydroxylation. This allows HIF-α to enter the nucleus, where it binds with HIF-1β to form the functional HIF complex, which then binds to HREs to promote the transcription of target genes involved in PHD regulation.

Under severe hypoxia, reduced oxygen levels inhibit PHD activity, preventing the degradation of HIF-α subunits (both HIF-1α and HIF-2α) [39]. Consequently, HIF-α escapes the ubiquitin–proteasome system (UPS), binds to the coactivator p300/CBP, and is transported into the nucleus to activate transcription with HIF-1β [40]. Typically, these genes enhance oxygen supply to tissues and facilitate metabolic adaptation to hypoxia. Despite their considerable similarities, HIF-1 and HIF-2 exhibit distinct functions and spatio-temporal regulations [41,42].

HIF-1α is rapidly activated during acute, severe hypoxia (1–2% O_2_). A higher prevalence of HIF-1α polymorphisms has been observed in PD patients, with some single nucleotide variants associated with an increased risk of developing PD (Figure 1) [43]. Hypoxia has been reported to upregulate the transcription and expression of tyrosine hydroxylase, the rate-limiting enzyme for dopamine synthesis, and the dopamine transporter (DAT), both crucial for dopaminergic neuronal function, via HIF-1α regulation [1,44]. Conditional knockout mice lacking HIF-1α exhibit reduced levels of vascular endothelial growth factor (VEGF) and TH in midbrain-derived neural precursor cells, leading to decreased differentiation of dopaminergic neurons [45,46,47]. This evidence indicates that HIF-1α plays a crucial role in the survival and functioning of dopaminergic neurons in the SNpc region.

Additionally, activation of HIF-1α is associated with suppressed mitochondrial function. HIF-1α negatively regulates mitochondrial function by repressing the transcriptional activity and inhibiting the activity of cellular myelocytomatosis viral oncogene in VHL-deficient renal cell carcinoma, which may contribute to mitochondrial dysfunction in PD [48]. However, the function of HIF-1α appears to be diminished in PD patients. Gene expression profiling analysis reveals decreased levels of HIF-1α and its target genes (including VEGF and hexokinase) in PD patients, with an upregulation of PHD2 in the SNpc homogenate of PD patients compared to age-matched controls.

The evidence suggests that HIF-1α exhibits promising neuroprotective functions against PD. However, paradoxical findings indicate that during severe hypoxia in the brain, HIF-1α may also facilitate inflammasome formation, mitochondrial dysfunction, and cell death [49]. For instance, in neuron-specific conditional knockout mice, deficiency of neuronal HIF-1α and HIF-2α improves neuronal survival in the early acute phase following ischemic stroke. Conversely, similar experimental designs have yielded different outcomes, with some studies showing that neuron-specific HIF-1α knockdown increases tissue damage and reduces survival rates in response to transient focal cerebral ischemia. These discrepancies might be due to variations in the murine stroke models used [50]. Additionally, HIF-1α regulates genes involved in autophagy and apoptosis, such as BNIP3 and Noxa (or PMAIP1, phorbol-12-myristate-13-acetate-induced protein 1) [51,52]. This adds an extra layer of complexity to its role in cellular regulation, making HIF-1α a potential double-edged sword in the context of hypoxia, oxidative stress, or ischemia. While moderate stress levels might activate the neuroprotective aspects of HIF-1α, even slight deviations could trigger detrimental effects [53]. These paradoxes highlight the intricate and sophisticated role of HIF-1α, which must be carefully considered in PD therapy development.

In contrast, HIF-2α gradually accumulates during prolonged, moderate hypoxia (<5% O_2_) and regulates similar genes [54,55]. Although the research on the correlation between HIF-2 and PD is limited, HIF-2 likely has functions that overlap significantly with those of HIF-1. This is suggested by the comparable upregulation of HIF-1-regulated genes observed in response to short hypoxic episodes in brain tissue from HIF-1 knockout mice [56]. In post-mortem PD brains, the accumulation of HIF-2α is observed as a marker of chronic hypoxia [57]. Additionally, the data suggest that HIF-2α promotes α-syn hyperphosphorylation at the serine 129 site (pS129-αSyn) and abnormal aggregation of pS129-αSyn by upregulating alkaline ceramidase 2 (Acer2), which leads to cognitive impairment in mice. This finding is consistent with elevated levels of pS129-αSyn in the plasma of patients with obstructive sleep apnea, a condition associated with chronic intermittent hypoxia [58]. Thus, like HIF-1α, HIF-2α also exhibits paradoxical effects, potentially being either neuroprotective or detrimental to neurons.

### 2.2. Nrf2/HO-1 Pathway

As a central regulator of cellular redox states, Nrf2 plays a crucial role in protecting cells against oxidative stress by activating the transcription of a wide range of antioxidant and anti-inflammatory genes. This regulation impacts metabolic pathways and initiates the production of NADPH and ATP [59]. Nrf2 is regulated by Kelch-like ECH-associated protein 1 (KEAP1), a substrate adaptor that forms a complex with Cullin-3 (CUL3) to target specific proteins for degradation. Under normal conditions, KEAP1 and CUL3 form a ubiquitin E3 ligase complex that polyubiquitinates Nrf2, leading to its rapid degradation via the proteasome system. Whether under normoxia or hypoxia, Nrf2 degradation through the UPS depends on direct interaction with KEAP1, but this degradation is more rapid in normoxia than in hypoxia [60], aligning with Nrf2’s role as a hypoxia regulator [61].

Under oxidative stress or specific conditions, Nrf2 translocates from the cytoplasm to the nucleus, where it forms a heterodimer with small MAF (sMAF) proteins and binds to the antioxidant response element (ARE) to activate a range of phase II detoxification genes, thus functioning as a major regulatory transcription factor (Figure 2). The beneficial role of Nrf2 in neurodegenerative conditions and the potential of specific Nrf2 activators as therapeutic agents are well-documented [62]. While the antioxidant functions of Nrf2 have been a primary focus, in vitro studies also show that Nrf2 accelerates the clearance of α-syn and mutant leucine-rich repeat kinase 2 (LRRK2) through the UPS [63]. As a transcription factor, Nrf2 exerts its neuroprotective effects by activating multiple protective genes involved in various molecular events of PD [64]. Among these genes, one notable candidate is heme oxygenase-1 (HO-1).

HO-1 is a potent antioxidant enzyme that degrades heme into carbon monoxide, free iron, and biliverdin [65]. The promoter region of the HO-1 gene contains various binding sites for transcription factors such as Nrf2 and HIF-1 [66]. Significantly elevated levels of HO-1 have been observed in the plasma of early-stage PD patients and in the cerebrospinal fluid of children following severe traumatic brain injury [67]. These observations suggest an underlying connection between HO-1 and PD.

Consistent with the findings on Nrf2, in vitro studies have shown that overexpression of HO-1 enhances the degradation of α-syn through the UPS. The α-syn mutant A30P exhibits resistance to HO-1-dependent degradation, leading to increased toxic intracellular aggregation. This finding suggests that Nrf2 may facilitate α-syn clearance through the Nrf2/HO-1 pathway [68]. Furthermore, HO-1 helps prevent dopaminergic neuronal death by promoting the expression of neurotrophic factors and enhancing the antioxidant response in both in vivo and in vitro PD models [69]. Chemicals that activate the Nrf2/HO-1 pathway have demonstrated therapeutic effects against PD models induced by 1-methyl-4-phenyl-1,2,3,6-tetrahydropyridine (MPTP), 1-Methyl-4-phenylpyridinium (MPP+), and other agents [70,71]. Nevertheless, the neuroprotective function of HO-1 appears to be closely associated with both the Nrf2 and HIF-1 pathways, highlighting the potential of targeting the Nrf2/HO-1 and HIF-1/HO-1 pathways for PD therapy.

## 3. The Correlation between Hypoxia and PD Pathogenesis

Hypoxia significantly affects neuronal health, particularly impacting dopaminergic neurons in the substantia nigra, which are crucial for motor control [72,73,74]. Hypoxia regulators and pathways play a key role in neuroprotection and in mitigating the progression of PD [75]. PD patients show a marked reduction in antioxidant defenses within the SNpc and lower levels of antioxidant proteins in the peripheral blood [76]. Drugs targeting hypoxia pathways have also shown great potential in alleviating PD symptoms [77,78]. Research on the relationship between hypoxia and PD has been conducted in the Han Chinese population [79]. Therefore, elucidating how PD development and symptoms interact with hypoxia pathways is crucial for understanding and potentially improving therapeutic strategies (Figure 3).

### 3.1. Dopaminergic Neurons’ Susceptibility to Hypoxia

Dopaminergic neurons in the SNpc, which innervate the striatum through long branching axons, are highly susceptible to hypoxia [80]. These neurons have high oxygen and ATP demands due to their extensive axonal projections, making them particularly vulnerable to oxidative stress and hypoxia [81,82]. The oxygen dependence of dopamine biosynthesis and metabolism exacerbates this vulnerability [83]. Additionally, dopamine metabolism is prone to generating oxidative stress, necessitating a robust antioxidant system [84]. The susceptibility of dopaminergic neurons to oxidative damage is further highlighted by their sensitivity to hypoxia and energy deficiency. Studies in rat brains have demonstrated that hypoxia or glucose deprivation can inhibit neuronal firing and induce hyperpolarization in dopaminergic neurons [85], indicating reduced neuronal excitability and activity.

### 3.2. Evidence of Hypoxia in PD

The evidence of hypoxia-related events in PD brains suggests that hypoxia may contribute to the pathogenesis of PD. In PD patients, the chemo sensitive receptors that control respiration by sensing blood pH fluctuations are impaired, leading to a diminished sensation of dyspnea and a reduced hypoxic response [86]. The inspiratory muscle strength is compromised even in very early-stage PD patients [87], and detrimental changes in lung volumes are observed, such as airflow limitations [88], restrictive patterns [89], or mixed patterns [90]. Acute levodopa administration has been found to improve inspiratory muscle function in both anesthetized dogs [91] and patients, and dopamine has been shown to enhance diaphragm function during acute respiratory failure in patients with chronic obstructive pulmonary disease [92]. These findings suggest that respiratory symptoms in PD may be related to the loss of dopaminergic neurons.

Furthermore, reduced respiratory functions, declines in autonomic regulation of the circulatory system, and inadequate cerebral perfusion could exacerbate the hypoxic environment in the brains of PD patients. At the molecular level, HIF-2α, a marker of chronic hypoxia, is elevated in the SNpc of post-mortem PD patients. Additionally, higher polymorphisms of HIF-1α have been observed in PD patients. Reduced levels of brain-derived neurotrophic factor (BDNF), which is important for synaptic development and plasticity [93], have also been noted in PD, schizophrenia, and Alzheimer’s disease [94]. The promoter region of the BDNF gene contains a bHLH-PAS transcription factor response element that interacts with HIF-1β to activate transcription by HIF [95]. Thus, decreased BDNF levels may indicate impaired hypoxia resistance in PD patients.

### 3.3. The Role of Hypoxia in PD Pathogenesis

Hypoxia disrupts normal mitochondrial function at the cellular level, leading to increased oxidative stress and decreased energy production. This disruption causes an accumulation of ROS, which further damages brain tissue and accelerates neurodegeneration. Dopaminergic neurons are particularly vulnerable to hypoxia due to their close association with mitochondrial function. Notably, the accumulation of ROS can result in two distinct outcomes: extreme hypoxia leads to significant oxidative stress, while moderate oxygen deprivation acts as a signaling mechanism that enhances cellular resilience [96].

### 3.4. The Role of Hypoxia in Other Neurodegenerative Diseases

Hypoxia may play a role in other neurodegenerative diseases as well. Indeed, hypoxia poses equivalent risks to the health of various neurons, not just dopaminergic ones. In tauopathies, hypoxic exposure is associated with increased activation of kinases that promote tau hyperphosphorylation, exacerbating tau pathology and facilitating the formation of protein aggregates, such as Aβ plaques and neurofibrillary tangles. This cascade results in oxidative stress, synaptic loss, and ultimately, cell death [97]. Both oxidative stress and tau hyperphosphorylation are crucial in the pathophysiology of tauopathies [98,99].

Atypical parkinsonisms include corticobasal degeneration (CBD), progressive supranuclear palsy (PSP), multiple system atrophy (MSA), and dementia with Lewy bodies (DLB), each characterized by different pathogenic proteins. MSA is a progressive and fatal form of atypical parkinsonism characterized by signs of chronic hypoxia in patients’ brains [100,101]. The research indicates that repeated episodes of cerebral perfusion followed by secondary chronic hypoxia can lead to increased vascular pathology and concurrent neurodegeneration, contributing to cognitive impairment in synucleinopathies [102].

## 4. The Mechanism of Hypoxia in Causing PD

Several theories regarding the pathogenesis of PD emphasize the roles of α-syn accumulation, mitochondrial dysfunction, lysosomal damage, and alterations in receptors/neurotransmitters in both sporadic and familial cases of PD. Hypoxia, a significant environmental factor, may influence the development of PD by affecting these pathways through various mechanisms. The interaction between these mechanisms forms a complex network, where they mutually influence and exacerbate each other throughout the pathological process of PD (Figure 4).

### 4.1. α-Syn Accumulation

The main pathological feature of PD is the degeneration and death of dopaminergic and other pigmented neurons in the SNpc. The link between α-syn and PD was first identified in 1997. In familial PD, six missense point mutations in the SNCA gene (also known as PARK1) promote the formation of α-syn aggregates and fibrils, thereby increasing the risk of PD [103].

α-syn is a soluble protein composed of 140 amino acid residues encoded by the SNCA gene, primarily found in the presynaptic nerve endings of the central nervous system. The protein consists of three regions: an amphiphilic helical structure at the N-terminal, a central hydrophobic region known as the NAC, which is critical for detecting α-syn aggregation, and a highly negatively charged C-terminal that forms an acidic, irregular curl structure [104,105]. α-syn has several physiological functions, including maintaining synaptic function [106], regulating dopamine and other neurotransmitter release [107], preserving cell membrane homeostasis [108], and influencing the production and function of microglia [109]. It also affects mitochondrial defects, autophagy, and lysosomal dysfunction [110].

Studies indicate that α-syn transmission relies on the release, internalization, and misfolding of α-syn [111]. Under physiological conditions, α-syn stabilizes into a tetramer of about 58 kDa [112]. However, in pathological states, α-syn misfolds, leading to abnormal aggregate formation [113]. The pathogenic amyloid aggregates formed by phosphorylated α-syn at tyrosine 39 (pY39α-syn) are highly cytotoxic to neurons and can induce more endogenous α-syn to undergo liquid–solid phase transformation, resulting in pathological protein aggregates [114].

The abnormal aggregation and transmission of α-syn play a crucial role in PD pathogenesis and are exacerbated under hypoxic conditions [18]. The research has shown that patients with obstructive sleep apnea syndrome have increased levels of total α-syn and phosphorylated α-syn (p-α-syn) in their plasma, which correlates positively with the oxygen desaturation index, suggesting that chronic intermittent hypoxia elevates α-syn levels [7]. During chronic hypoxia, hyperphosphorylation of α-syn occurs, enhancing ceramide catabolism through HIF-2α and HIF-2α-dependent transcriptional activation of Acer2, which is implicated in α-syn pathology and cognitive impairment in mice [58]. Additionally, exogenous α-syn oligomers promote HIF-1α accumulation in normoxic primary microglia via TLR7/8 mediation, thereby enhancing their migratory capacity [115]. The α-syn/HIF-1α axis induces neuroinflammatory injury through the IL6ST-AS/STAT3/HIF-1α axis. Inhibition of this axis in microglia, induced by α-syn stimulation, reduces HIF-1α expression and increases oxidative stress injury in SH-SY5Y cells [116]. Hypoxia/ischemia-related increases in α-syn levels may contribute to protein aggregation, and hypoxia leads to toxic α-syn oligomer formation within cells [27,117]. Moreover, abnormal α-syn aggregation further impairs the function of the ubiquitin–proteasome system, which mediates HIF-1α degradation [2,118].

This connection between hypoxia/ischemia and PD is further supported by other evidence. The intestine is considered a possible origin of α-syn pathology, with hypoxia and hypoxia signaling pathways playing significant roles in intestinal diseases. Chronic intestinal inflammation can also drive PD progression [119], with a clear mutual promotion between hypoxia and inflammation, the latter being a critical factor in promoting α-syn aggregation and dissemination [60]. However, specific studies linking hypoxia to inflammation-driven α-syn aggregation and dissemination are lacking.

Calcium ions are important signaling molecules in cells. Studies have shown that in human cell lines expressing α-syn, transient intracellular calcium increases lead to higher α-syn aggregate levels [120]. Calcium-binding protein calmodulin alters its conformation and binds to α-syn, thereby inducing α-syn fibrillation [121]. Additionally, when intracellular calcium concentration rises, the Ca^2+^/calmodulin complex can stabilize HIF-1α, enhancing its transcriptional activity [122]. Preventing iron accumulation is associated with α-syn aggregation and dopaminergic neuron vulnerability [123,124]. HIF promotes iron transport in the blood and iron absorption by hematopoietic cells, providing a theoretical basis for the positive effect of HIF-1α stabilization in PD treatment [125].

Hypoxia may trigger, promote, and exacerbate the abnormal pathology of α-syn through these mechanisms, though the underlying processes remain unclear. Additionally, there is currently no research indicating that hypoxia has a more profound impact on PD, highlighting the need for further investigation.

### 4.2. Mitochondrion Dysfunction

Mitochondria are critical organelles within cells. The introduction of toxins like MPTP inhibits mitochondrial complex I and disrupts the mitochondrial ETC, leading to severe degeneration of dopaminergic neurons in the substantia nigra and striatum. This process induces an acute and irreversible PD-like syndrome, highlighting the importance of mitochondrial function in PD research [73]. Increasing evidence suggests that PD is characterized by mitochondrial defects, including reduced mitochondrial numbers, structural damage, and decreased activity of respiratory chain complex I. Sporadic Parkinson’s disease, which accounts for over 90% of PD cases, is associated with disruptions in pathways that regulate mitochondrial energy production in nerve cells [126].

Although there is no direct evidence that PD-related genes cause hypoxia, mitochondrial dysfunction is a key mechanism through which these genes contribute to the disease, leading to oxygen utilization disorders. Hypoxic conditions can alter mitochondrial membrane structure and interactions with α-syn, potentially exacerbating Lewy body pathology and mitochondrial dysfunction. Therefore, hypoxia-induced mitochondrial dysfunction may interact with α-synuclein pathology [2]. Overexpression of α-syn can cause mitochondrial rupture, changes in mitochondrial membrane permeability, and decreased respiratory function, ultimately leading to neuronal death [127,128]. Additionally, α-syn can impair autophagy, including mitophagy, preventing the clearance of dysfunctional mitochondria and further exacerbating damage. The heterozygous GBA1 mutation, a common genetic risk factor for PD, leads to α-syn aggregation and mitochondrial dysfunction, increasing PD risk by more than 20 times [129].

ROS plays a crucial role in the interplay between PD and hypoxia. Mitochondria are a primary source of intracellular ROS, and elevated ROS levels can cause oxidative stress, leading to severe cellular damage [130]. This imbalance can result in the oxidative modification of α-syn [131], as well as lipid peroxidation, protein oxidation, and DNA damage, contributing to the loss of dopaminergic cells in the substantia nigra, a hallmark of PD [132].

Hypoxia can increase mitochondrial ROS production by limiting oxygen availability to complex III and regulating antioxidant gene expression [1]. Activation of the HIF-1 signaling pathway inhibits the expression of key mitochondrial ribosome components and ETC proteins COX2 and CYTB, resulting in abnormal mitochondrial membrane potential [133]. ROS can increase HIF-1α stability, while HIF-1α helps mitigate excessive ROS production by modulating mitochondrial oxygen consumption and inhibiting mtDNA-encoded mRNA expression [1]. Compared with wild-type cells, HIF-1α-knockout cells produce more ROS due to increased ATP production and impaired mitophagy [134]. The data show that arsenide stabilizes HIF-1α in a ROS-dependent manner by consuming ascorbic acid and Fe (II), which inhibits PHD activity and contributes to HIF-1α stability [135]. ROS-induced attenuation of NO-mediated HIF-1α regulation is mediated through the active proteasome degradation pathway [136]. Inhibition of mitochondrial ROS can eliminate Ang II-induced HIF-1 protein stability, HIF-1-dependent transcription, and VSMC migration [137]. ROS also influences the regulation of HIF-1-activated VEGF [138] and upregulates HIF-1α expression through the PI3K/AKT signaling pathway [139].

Notably, ROS activates Nrf2, another key hypoxia regulator. Knockdown of Nrf2 results in decreased antioxidant gene expression and increased ROS levels [140]. The Keap1-Nrf2 pathway regulates mitochondrial and cytoplasmic ROS production through NADPH oxidase [141]. Interestingly, while hypoxia-induced mitochondrial ROS formation stabilizes HIF-1α [142], HIF-1α is also stabilized in cells lacking mitochondrial respiration (U0 cells) [143]. Further research is needed to explore how factors such as hypoxia duration and oxygen concentration may influence these interactions.

### 4.3. Lysosome Damage

Mitochondrial dysfunction can lead to lysosomal damage [144,145,146]. Lysosomes play a crucial role in supporting mitochondrial metabolism by regulating iron homeostasis [147,148]. Dysfunctional lysosomes can impair mitochondrial function, leading to the accumulation of damaged mitochondria [149]. The physical and functional communication between mitochondria and lysosomes underscores this interaction.

Under hypoxic conditions, mitochondrial–lysosomal contact is enhanced, with some lysosomes being engulfed by enlarged mitochondria, a process termed giant mitochondrial engulfing lysosomes (MMEL). Following MMEL, the lysosomal membrane ruptures, releasing lysosomal proteases into the enlarged mitochondria. This process, combined with mitochondrial proteases, facilitates mitochondrial self-digestion under hypoxia and promotes mitochondrial ROS production [150].

Lysosomes are central to cellular homeostasis, managing the turnover of proteins, lipids, and other macromolecules through endocytosis, phagocytosis, and autophagy [151]. In neurodegenerative diseases like PD, frontotemporal dementia, and ALS, numerous risk variants in lysosomal genes have been identified [152,153]. Dysfunction in the lysosomes of glial and neuronal cells can lead to the spread of pathological features, such as amyloid-β and tau in AD and α-syn in PD [154,155,156].

Hypoxia can alter the distribution of lysosomal proteins, resulting in lysosomal membrane permeability [157]. The ATPase H^+^ transporter V1 subunit A (ATP6V1A) is vital for maintaining lysosomal homeostasis. Hypoxia impairs lysosomal function in head and neck squamous cell carcinoma by binding HIF-1α to the E-box motif in the ATP6V1A promoter, leading to downregulation of ATP6V1A expression [158]. This impairment reduces the fusion of multivesicular bodies (MVBs) with lysosomes and decreases the secretion of intracavitary vesicles (ILVs) into extracellular vesicles (EVs). Hypoxia disrupts lysosomal function by affecting lysosomal acidification rather than its biosynthesis. Re-expressing ATP6V1A under hypoxic conditions can restore lysosomal function and EVs release levels.

Furthermore, overexpression of HIF-1α and inhibition of HIF-1α ubiquitination impair lysosomal function and enhance EV secretion. Several studies have documented the impact of HIF-1α on EV release [159,160]. For instance, King et al. reported that increased expression of plasma membrane receptors regulated by HIF-1α can lead to self-activation, promoting endocytosis and EV release. Studies have also shown that lysosomal-associated membrane proteins (LAMPs) and the key lysosomal protease cathepsin D (CTSD) are downregulated in hypoxic-exposed trophoblast cells, eventually leading to lysosomal dysfunction [161]. In addition, hypoxia and the resulting acidosis in tumors can cause lysosomal movement, abnormal expression of LAMPs, lysosomal compartment expansion, and lysosomal deacidification, all associated with impaired lysosomal activity [162,163,164,165].

Under hypoxic conditions, chondrocytes exhibit increased levels of the autophagy marker LC3, the lysosomal marker LAMP1, and CTSD. This results in an increase in the number of autophagosomes, lysosomes, and defective lysosomes. Transcription factor EB, a key regulator of lysosomal biosynthesis, has transcription binding sites in its promoter region regulated by HIF-1α and HIF-2α, indicating that HIF-α is involved in regulating lysosomal biosynthesis and function. Additionally, hypoxia increases endoplasmic reticulum (ER) stress in chondrocytes via the PERK signaling pathway, which is upstream of the autophagy–lysosomal signaling pathway [166]. Hypoxia can also enhance ROS elimination by activating autophagy, thereby increasing cellular resistance to radiation [167].

### 4.4. PD and Receptors/Neurotransmitters

Hypoxia and neurotoxicity are potential contributors to PD, with δ-opioid receptors (DOR) showing neuroprotective effects against hypoxia and ischemic injury. In the brain, MPTP is metabolized by monoamine oxidase B (MAO-B) in glial cells into an intermediate called MPDP+. This intermediate is further oxidized to form the toxic MPP+. MPP+ is a neurotoxin that mimics PD by disrupting dopaminergic neurons in the substantia nigra both in vivo and in vitro [168,169]. In SH-SY5Y PD model cells, the inhibition of complex I by rotenone or MPP+ also results in decreased levels of HIF-1α [170].

Both hypoxia and MPP+ stress have been identified as potential pathogenic factors leading to PD [85,171]. Prolonged hypoxia or high concentrations of MPP+ reduce cell viability, decrease PINK1 protein levels, and increase caspase 3 cleavage. DOR activation protects cells from hypoxia and MPP+ injury by upregulating PINK1 and downregulating cleaved caspase 3. In contrast, the inhibition of DOR exacerbates these effects.

Previous studies have demonstrated that they induce apoptosis by disrupting mitochondrial stability [172,173,174]. Additionally, apoptosis has been found to be caspase-dependent [175]. Activation of DOR by UFP-512 significantly reduces hypoxia-induced cell damage and α-syn overexpression. DOR antagonists like naltrindole or DOR siRNA knockdown reduce hypoxia-induced cell damage and α-syn overexpression. These findings suggest that DOR activation mitigates MPP+ or severe hypoxia-induced aggregation of α-syn through the CREB pathway [27]. However, unlike MPP+, hypoxia-induced mitophagy is not dependent on DOR-PINK1 signaling pathway [176]. Additionally, in PD models, changes in the serotonin system, beyond the dopamine pathway, have been noted [177,178,179].

5-Hydroxytryptamine (5-HT), a monoamine neurotransmitter, regulates hypoxic ventilatory responses. The 5-HT 2 receptor is involved in both normoxic and hypoxic respiration in PD model rats. A unilateral injection of 6-hydroxydopamine (6-OHDA) into the rat medial forebrain bundle (MFB) affects the hypoxic ventilation response via 5-HT 2 receptor stimulation [180]. However, some studies suggest that the raphe in the medulla oblongata and its 5-HT neurons do not mediate the respiratory response to hypoxia, and 5-HT absence in transgenic mice does not affect long-term ventilation after intermittent hypoxia [181]. Patients with mild PD exhibit weakened hyperventilation responses to hypoxic stimulation. In the rat MFB PD model, dopamine depletion primarily leads to enhanced ventilation responses to hypoxia and the disappearance of central and peripheral D2 dopamine receptor antagonism on respiration under both hypoxic and normoxic conditions [180].

DJ-1 plays a crucial role in the expression of rearrangement (RET), a receptor for neurotrophic factors and a neuroprotective molecule for dopaminergic neurons. Its ligand, glial-cell-line-derived neurotrophic factor, promotes the survival and differentiation of midbrain DA cells, making RET a key target in PD degeneration [182]. The inducible loss of DJ-1 triggers hypoxia and ROS production, stabilizing HIF-1α. HIF-1α expression is essential for RET downregulation [183].

The synthesis of neurotransmitters and modulators is regulated by oxygen-demand rate-limiting enzymes, meaning hypoxia, which disrupts O2 homeostasis, can affect neuronal function by altering neurotransmitter synthesis. Hypoxia can be classified into persistent hypoxia and intermittent hypoxia [184]. Dysfunction of the nicotinic acetylcholine receptor (nAChR) is associated with PD. In addition, HIF-1α also interacts with mitochondrial nAChRs. Nicotine stimulates HIF-1α expression via activation of α7 nAChRs, which interact with HIF-1α to regulate its translocation to the nucleus and mitochondria [185]. Nitric oxide (NO) is involved in neurotransmitter-mediated activation and inhibition of the inflammatory cascade. Excessive NO production by activated microglia is associated with neuroinflammation and neurodegenerative diseases. Short-term inhibition of neuronal nitric oxide synthase (nNOS) by CH is due to reduced oxygen supply [186].

As mentioned, molecular events specific to PD pathogenesis can interfere with the expression and stability of HIF-1α. These disruptions can impair cellular antioxidant capacity, mitochondrial function, and metabolic processes. HIF-1α activates several transcriptional processes targeting oxidative stress, including autophagy and mitochondrial function, that influence PD development [1]. It is important to note that the pathogenesis of PD remains unclear and may involve various molecular events, each affecting hypoxic pathways and treatment responses differently.

## 5. The Role of PD-Related Genes in Hypoxia

To date, over 20 genetic variations associated with PD have been identified, collectively accounting for about 5–10% of PD cases. Among these, mutations in genes encoding the proteins Parkin, PINK1, DJ-1, and TMEM175 can lead to autosomal recessive PD. These genetic mutations are closely related to PD risk and provide significant insights into its pathogenesis. Notably, several of these genes are linked to mitochondrial dysfunction and hypoxic responses (Figure 5) [123].

### 5.1. Hypoxia Promotes Parkin/PINK1 Pathway

Parkin (PARK2) is a RING-in-between-RING (RBR) type E3 ubiquitin ligase. The PARK2 mutation is the most common cause of recessive PD, particularly in familial early-onset cases, accounting for over 70% [187]. Mutations in Parkin increase oxidative stress, promote mitochondrial swelling and fission, and affect mitochondrial phagocytosis, leading to mitochondrial defects and, ultimately, resulting in the loss of dopamine neurons [188,189]. Parkin is involved in ubiquitination, and mutations disrupt its E3 ligase activity, including its binding to α-syn, which contributes to the formation of Lewy bodies [190]. Interestingly, recent studies using quantitative proteomics have identified HIF-1α as a potential ubiquitination substrate for Parkin [191]. Parkin has been shown to reduce HIF-1α levels in glioblastoma cells. It binds to HIF-1α and promotes HIF-1α degradation through ubiquitination, thereby inhibiting breast cancer cell metastasis. This effect is independent from oxygen levels [192].

HIF-1α, in turn, promotes Parkin/PINK1-mediated mitophagy [193] and can upregulate mitophagy in the spinal cord of mice with diabetic neuropathic pain through the Parkin signaling pathway [194]. Additionally, FSH activates mitophagy via the HIF-1α-PINK1-Parkin pathway, reducing hypoxia-induced apoptosis in porcine ovarian granulosa cells [195]. Parkin regulates the expression of HIF-1α and HIF-3α in glioblastoma-derived cell lines in vitro. Under normoxic conditions, Parkin silencing leads to HIF-1α accumulation in the nucleus, while under hypoxic conditions, Parkin affects HIF expression. Parkin knockdown decreases HIF-3α immunoreactivity in both the cytoplasm and nucleus, suggesting a positive role for Parkin in regulating HIF-3α [196]. Conversely, the inhibitory PAS domain protein (IPAS), a splicing variant of HIF-3α, generally inhibits HIF-1, and reduced PINK1-Parkin activity stabilizes IPAS, increasing its mitochondrial level and affecting HIF-1α expression [197]. The relationship between Parkin and hypoxia pathways warrants further investigation.

PTEN-induced putative kinase 1 (PINK1) is a serine/threonine kinase located in mitochondria and cytoplasm, crucial for regulating mitochondrial function. By phosphorylating mitochondrial proteins, PINK1 helps protect against mitochondrial dysfunction during cell stress [198]. PINK1 gene knockout leads to mitochondrial dysfunction and potential dopaminergic neuron death. Mutations in PINK1 account for about 9% of familial early-onset PD cases.

PINK1 is essential for the translocation of Parkin to depolarized mitochondria, promoting the clearance of Parkin through mitophagy [199]. PINK1 recruits Parkin from the cytoplasm to mitochondria, where Parkin facilitates mitochondrial phagocytosis [200]. Studies have demonstrated a strong correlation between PINK1 and HIF-1α. PINK1 knockout alters the expression of many genes in the HIF-1α pathway [201]. DOR protects neurons from hypoxia and MPP+ by upregulating PINK1 [202].

Using hypoxic stress models, researchers have found that PINK1 deficiency attenuates the induction of HIF-1α across three different cell types. PINK1 mediates the HRE gene response required for handling hypoxic stress response through HIF-1α translation regulation [203]. This suggests that changes in PINK1 expression may be an adaptive response to hypoxic environments. Under hypoxic conditions, PINK1 knockout mice show significantly reduced HIF-1α protein accumulation, transcriptional activity, and expression of hypoxia-responsive genes [204]. HIF-1α mitigates oxidative stress through the HEY1/PINK1 pathway, and PINK1 expression can be modulated by changes in oxygen levels, impacting mitochondrial development in tumor cell lines [205]. Additionally, the JCYSTL formula activates PINK1/Parkin-mediated mitophagy by stabilizing HIF-1α [206]. Given the close interaction between Parkin and PINK1 in mitochondrial regulation, the links between PINK1 and HIF-1α are also likely involved in Parkin’s function.

### 5.2. DJ-1 Reduces Oxidative Stress

DJ-1 (PARK7) has over 20 mutations associated with early-onset PD [207]. It mitigates oxidative stress through various signaling pathways, serving as a cellular protective mechanism [208]. In the human brain, DJ-1 is highly expressed in reactive astrocytes but less so in neurons [209]. DJ-1 is crucial for mitochondrial function and acts as a redox sensor within mitochondria. Abnormal DJ-1 expression leads to mitochondrial defects and increased oxidative stress, affecting mitochondrial phagocytosis [210]. DJ-1 is involved in the hypoxia pathway and can be induced by oxidative stress, mediating hypoxia-induced cell responses.

As early as 1997, researchers identified DJ-1’s role in protecting neurons from hypoxia-induced cell death [211]. DJ-1 is essential for HIF-1α target gene expression. In U2OS cells, DJ-1 deficiency alters HIF-1α-induced gene expression and impairs transcription of many HIF-1α target genes. Additionally, DJ-1 is crucial for maintaining HIF-1α-stable Akt and mTOR activity [212]. DJ-1 deficiency reduces HIF-1α expression in neuroblastoma cells, and its regulation of the DJ-1/PI3K/AKT pathway impacts HIF-1α protein expression and transcriptional activity [213]. DJ-1 inhibits the VHL ubiquitination activity of HIF-1α subunit by inhibiting HIF-1α-VHL interaction. The decreased expression of VEGF mRNA after DJ-1 deletion suggests that DJ-1 mediates neuronal survival through the VHL-HIF-1α pathway [214].

### 5.3. LRRK2 Can Be Activated by HIF-1α

LRRK2, also known as PARK8, is widely expressed throughout the brain and surrounding tissues, where it influences neurite growth and autophagic protein degradation [215]. LRRK2 plays a role in mitochondrial function and cytoskeleton dynamics [216], which are critical for various cellular processes. Mutations in LRRK2 are a major cause of autosomal dominant PD and are associated with either mitochondrial phagocytosis and other normal function or α-syn pathology and other neurodegenerative proteins [217]. After traumatic brain injury, HIF-1α directly binds to the LRRK2 proximal promoter, leading to increased LRRK2 transcription and exacerbating neuronal cell death [218]. In cases of anemia or hypoxia, HIF binds to the erythropoietin (EPO) 5’ hypoxia response element, resulting in increased EPO gene transcription [219]. In the liver of LRRK2 knockout mice with hemangiomas, there is an increase in HIF-2α protein expression and significant reactivation of HIF-2α target gene EPO, reflecting the role of the HIF-2α pathway in angiogenesis [220]. Additionally, in human cancers, LRRK2 phosphorylates HIF-1α at Ser797, which enhances HIF-1α‘s binding affinity to its cofactor p300, thereby increasing the production of HIF-1α target genes.

### 5.4. TMEM175 May Influence HIF-1α

Lysosomal K+ conductance, mediated by the lysosomal membrane protein transmembrane protein 175 (TMEM175), is essential for maintaining lysosomal membrane potential, pH stability, and regulating lysosomal–autophagosome fusion [221,222]. TMEM175 acts as a proton-permeable, cation-selective channel, helping to stabilize intracellular pH by facilitating lysosomal hydrogen ion efflux. Compared to wild-type neurons, TMEM175 knockout neurons exhibit increased aggregation of p-α-syn, while TMEM175 overexpression reduces these inclusions, suggesting that TMEM175 enhances the lysosomal degradation of α-syn aggregates.

TMEM175 knockdown decreases Cathepsin B and Glucosylceramidase Beta (GBA) enzyme activity and promotes α-syn accumulation [223]. In mice, the loss of TMEM175 function results in dopaminergic neuron loss and progressive dyskinesia [224]. Dysfunction in mitochondrial autophagy caused by TMEM175 loss-of-function mutations is linked to several central nervous system diseases, including Alzheimer’s [225] and Parkinson’s diseases [226]. Additionally, TMEM175 protein levels decreased after cerebral ischemia–reperfusion (I/R) injury, and its overexpression can mitigate brain cell death and neurobehavioral deficits caused by cerebral artery occlusion/reperfusion in vivo [227]. Excessive NO production during cerebral I/R injury leads to upregulation of HIF-1α, triggering an inflammatory response and, ultimately, neuronal apoptosis, which may explain TMEM175’s effect on HIF-1α.

The pathogenic genes associated with PD discussed above are predominantly linked to hypoxic pathways and HIF-1α. They primarily influence PD progression by modulating cellular antioxidant capacity, mitochondrial function, autophagy pathways, and the aggregation of α-syn proteins.

## 6. Hypoxia-Based Therapeutic Strategies for PD

Currently, there is no cure for PD, but symptoms can be managed with medication. Approved treatments include DOPA decarboxylase inhibitors/replenishing lost dopamine (DA) precursors, COMT inhibitors, DA agonists, MAO-B inhibitors, and other drugs with mixed mechanism [228]. These medications primarily address PD symptoms by dopamine and alleviating DA deficiency. Since hypoxia signaling is intrinsically involved in the pathogenesis of PD, targeting the HIF and Nrf2 pathways has been proposed as a strategy to ameliorate or prevent PD [229,230]. Hypoxia pathways, particularly HIF-1α and Nrf2, are considered promising targets due to their connection with PD-related molecular events, including protein degradation, mitochondrial function, ROS generation, and genetic mutations [1]. Consequently, an increasing number of studies are investigating drugs that target HIF-1α and Nrf2, as well as treatments aimed at improving hypoxia, as novel approaches to modulate aberrant pathways and treat PD [231].

### 6.1. Drugs Targeting Hypoxia Pathways

Deferoxamine mesylate (DFO) is an iron chelator that also functions as a hypoxia-mimetic agent, often referred to as a HIF-1α activator [177]. DFO stabilizes HIF-1α by inhibiting PHD activity [232]. Additionally, DFO alleviates iron imbalances induced by cypermethrin in rats and reduces α-syn A53T aggregation in PC12 cells [233,234]. In SH-SY5Y cells, DFO treatment decreases apoptosis and ROS generation caused by H_2_O_2_ or 6-OHDA, a compound that selectively destroys dopaminergic neurons [235].

Furthermore, DFO enhances autophagy in SH-SY5Y cells treated with rotenone or MPP+ through the HIF-1α and Beclin-1 pathways [182]. Notably, DFO also promotes AMPK-mediated Nrf2 pathway activation by inhibiting ferroptosis, which leads to reduced ROS generation [236,237]. Other iron chelators, including M30, lactoferrin, clioquinol, and FG-0041, also block PHD activity through iron chelation and subsequently activate HIF-1α and potentially Nrf2, thereby enhancing neuronal function and viability [238,239].

Albendazole (ABZ), a benzimidazole anthelmintic, enhances the expression of HIF-1α and its downstream target, VEGF, while inhibiting PHD activity [240]. It significantly reduces histological alterations in the SNpc, restores DA levels and motor functions, and decreases α-syn expression in a rat model of PD induced by rotenone [241]. Additionally, ABZ increases the expression of nuclear receptor-related 1 (Nurr1) in the SNpc and enhances the transcriptional activation of Nurr1-controlled genes, which are crucial for regulating DA synthesis [241].

Agmatine, a natural metabolite of the amino acid arginine, activates HIF-1α and thus prevents ROS production, mitochondrial membrane potential loss, and increases in caspase 3 and cytochrome c in rotenone-treated differentiated SH-SY5Y cells [242]. Consequently, agmatine alleviates motor symptoms in PD rats, inhibits inflammatory and oxidative stress pathways, and increases striatal dopamine levels through the activation of Nrf2 and other pathways [243,244].

### 6.2. Behavioral Modulation Improving Hypoxia

Preclinical and clinical evidence suggest that regular exposure to intermittent hypoxia may benefit PD patients both in the short and long term, highlighting potential treatment targets [9,18,245]. One study involving a 7-day walk in the Swedish mountains demonstrated significant improvements in motor performance in PD patients, with effects observed 4 days post-intervention and lasting for 3 months, although not beyond 6 months [246]. Quality-of-life assessments showed improvements 1 week after the intervention, but these benefits did not persist for 18 weeks [247,248]. Additionally, negative feelings related to daily life appeared to be alleviated following the intervention [249].

Another study used a 14-day hypoxia protocol involving controlled oxygen levels of 8–10% for 5–7 minutes, three times per day, to investigate the interaction between DA metabolism and hypoxia-induced ventilatory responses (HVRs) [250]. This intervention significantly improved HVRs in the PD group, and DA metabolism also showed improvement compared to healthy controls [251]. These findings suggest that hypoxic training can enhance dopamine metabolism, HVRs, motor symptoms, and overall quality of life in PD patients. Consistent with these results, intensive exercise has been shown to improve motor and cognitive symptoms in rat models of PD and to restore striatal synaptic plasticity damaged by α-syn aggregation [252].

## 7. Conclusions and Perspectives

As the second most common neurodegenerative disease, PD imposes a significant burden on society, which is increasing due to the rising life expectancy of the population. Understanding the pathogenesis and progression of PD is, therefore, of great concern. There is growing evidence that the response to changes in oxygen levels is critical in neurodegeneration, particularly in PD. Hypoxia, which can act both as a cause and a consequence of PD symptoms, exacerbates the disease. Additionally, hypoxia is present throughout the progression of PD, significantly influencing its key aspects. Hypoxic insults can directly induce PD-like features, including motor disorders, mood changes, and brain lesions in susceptible individuals. Conversely, PD patients exhibit impaired responses to hypoxic stimuli, reduced ventilatory function, and diminished recovery following cumulative hypoxia [80].

These observations suggest that hypoxia pathways, particularly the HIF-1α and Nrf2 pathways, are intrinsically linked to PD-related molecular events. Various molecular changes specific to PD influence HIF-1α and Nrf2 in different ways [24,62]: some induce ROS and other stresses, leading to the activation of HIF-1α and Nrf2, while others accelerate their degradation or inhibit their functions, impairing the neurons’ ability to adapt to hypoxia. Over-activation of HIF-1α and Nrf2 pathways can lead to apoptosis, whereas reduced activity of these pathways increases vulnerability to oxidative stress, iron dysregulation, and protein misfolding in neurons [230].

The pathogenesis of PD remains incompletely understood, involving complex interactions among cellular, environmental, and genetic factors. Researchers have proposed various hypotheses to explain the onset and progression of the disease. Key theories suggest that neuronal degeneration is linked to abnormal accumulation of α-syn, mitochondrial dysfunction, oxidative stress, free radicals, and neuroinflammation [253,254]. Hypoxia plays a significant role in multiple stages of PD pathogenesis, including abnormal α-syn accumulation, mitochondrial dysfunction, lysosomal damage, and alterations in receptors and neurotransmitters. Furthermore, PD-related genes are closely linked to hypoxia pathways. This connection is strongly supported by numerous studies. Given the critical role of hypoxia in various stages of PD progression, targeting hypoxia pathways emerges as a promising strategy for developing new therapies for PD.

However, it is crucial to recognize that hypoxia plays a dual role in PD, influencing various aspects of the disease’s etiology. On one hand, hypoxia-induced neurotoxicity highlights the detrimental effects of low oxygen levels on neuronal health, contributing to brain damage and the onset of neurodegenerative disorders like PD [255]. As previously noted, hypoxia is a common feature in many neurological disorders, including PD [256]. For PD patients, particularly those with obstructive sleep apnea syndrome, hypoxia is associated with an increased risk of disease progression [7]. On the other hand, HIF-1α plays a unique neuroprotective role in PD. Insufficient levels of HIF-1α have been linked to the development of PD, underscoring the importance of this transcription factor in promoting cell survival, especially in dopaminergic neurons affected by PD. Additionally, δ-opioid receptor activation, as a HIF-1α stabilizer, has been shown to prevent hypoxia-related mitochondrial dysfunction associated with PD, suggesting a potential therapeutic target for mitigating the disease’s pathology under low oxygen conditions [6].

Recent studies on PD increasingly focus on the hypoxia pathway’s key protein, HIF-1α, which plays a crucial role in microglial immune memory and contributes to neuroinflammation in PD [257]. Additionally, α-syn induces neuroinflammation via the IL6ST-AS/STAT3/HIF-1α axis, providing a novel inflammatory perspective on the disease’s pathogenesis [116]. FG-4592, an inhibitor of HIF prolyl hydroxylases (HIF-PH), has been used to mitigate oxidative stress caused by α-syn, and its potential for drug repurposing offers new therapeutic avenues for PD [258]. Furthermore, researchers have identified a mitochondria-targeted iron chelator that protects dopaminergic neurons from mitochondrial toxins by modulating the HIF-1α pathway and reducing oxidative stress, underscoring its therapeutic potential [259].

Admittedly, the research on the relationship between hypoxia and PD still faces several debates and unanswered questions due to the complex and multifaceted effects of hypoxia on the disease. As mentioned before, the relationship between hypoxia and the onset and progression of neurodegenerative disorders continues to be debated [6]. Additionally, while mitochondria play a crucial role in regulating HIF-1α expression under hypoxic conditions, the mechanisms governing this regulation remain controversial [260]. Furthermore, studies indicate that activated hypoxia response pathways may exert varying effects across the spectrum of PD. There is ongoing debate about the optimal doses of hypoxic adaptation therapies and their potential risks [18]. Thus, there are unresolved issues regarding the effects of hypoxia on PD and other neurodegenerative conditions, as well as the diverse impacts of hypoxia response pathways and the regulatory mechanisms of HIF-1α in hypoxic environments.

To gain a clearer understanding of the connection between hypoxia and PD and to develop effective treatment strategies, further research is essential. The reciprocal relationship between hypoxia and PD underscores the need to balance the harmful effects of low oxygen on neuronal health with the potential neuroprotective mechanisms that could mitigate the disease’s pathology. Future studies on the role of hypoxia in PD should focus on the effectiveness and underlying molecular mechanisms of hypoxia-based neuroprotective strategies [101]. Investigating the connections between hypoxia and PD pathophysiology—such as its effects on α-syn pathology, mitochondrial dysfunction, and the vulnerability of dopaminergic neurons—could provide crucial insights for developing novel treatment approaches. Additionally, research should explore the potential benefits of hypoxia-based therapy protocols, including continuous and intermittent hypoxia, with a focus on specific treatment plans and outcomes for PD patients [261]. Addressing these research gaps will enhance our understanding of hypoxia’s role in PD progression and may lead to the development of innovative therapeutic strategies.

## Figures and Tables

**Figure 1 ijms-25-10484-f001:**
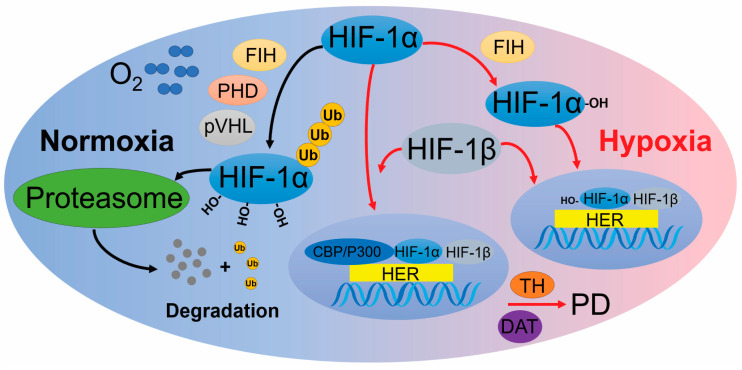
The HIF pathway in hypoxia. Under conditions of sufficient oxygen, PHD and FIH hydroxylate HIF-1α. Hydroxylated HIF-1α is subsequently recognized by VHL, leading to its ubiquitination and degradation. As a result, the concentration of HIF-1α remains very low. In moderate hypoxia, only FIH hydroxylates HIF-1α, allowing HIF-1α to translocate into the nucleus, where it binds with HIF-1β to form the HIF complex. This complex then binds to HREs in the genome, initiating transcription of target genes. In severe hypoxia, unhydroxylated HIF-1α not only enters the nucleus and binds to HIF-1β but also interacts with the coactivator p300. This interaction enhances the transcription of downstream regulatory genes. Abbreviations: PD: Parkinson’s disease; HIF-1α: hypoxia inducing factor-1α; FIH: factor inhibiting HIF; PHD: hypoxia-inducible factor prolyl hydroxylase; pVHL: von Hippel–Lindau protein; Ub: ubiquitin; Proteasome: 26S proteasome; HIF-1β: hypoxia inducing factor-1β; CBP/P300: the coactivator CBP/P300; HER: hypoxia response elements; TH: tyrosine hydroxylase; DAT: dopamine transporter; O_2_: oxygen.

**Figure 2 ijms-25-10484-f002:**
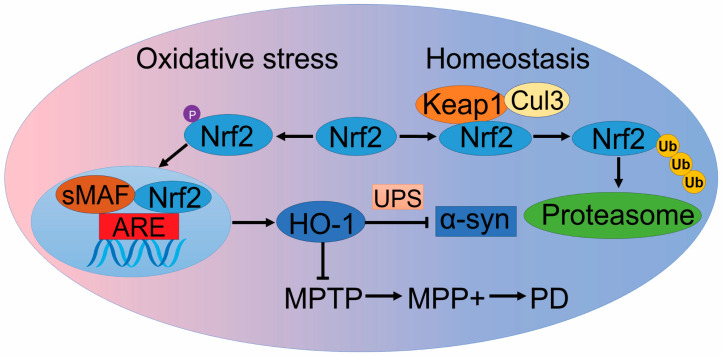
The Nrf2/HO-1 pathway in hypoxia. In the cytoplasm, KEAP1 forms a ubiquitin E3 ligase complex with CUL3 to polyubiquitinate NRF2, leading to its rapid degradation by the proteasomal system. Under oxidative stress, NRF2 is released from KEAP1-mediated inhibition. NRF2 translocates to the nucleus, where it heterodimerizes with the small Maf protein. Then, this NRF2-Maf complex binds to the ARE in the genome, allowing the expression of a series of downstream protective phase II detoxification enzyme and antioxidant enzyme genes and proteins, such as HO-1. Abbreviations: Nrf2: nuclear factor erythroid 2-related factor 2; sMAF: myeloma-associated factors; ARE: antioxidant response element; HO-1: heme oxygenase-1; UPS: ubiquitin proteasome system; MPTP/MPP+: 1-methyl-4-phenyl-1,2,3,6-tetrahydropyridine/1-Methyl-4-phenylpyridinium; α-syn: α-synuclein; Keap1: Kelch-like ECH-associated protein 1; Cul3: Cullin-3; Ub: ubiquitin; Proteasome: 26S proteasome.

**Figure 3 ijms-25-10484-f003:**
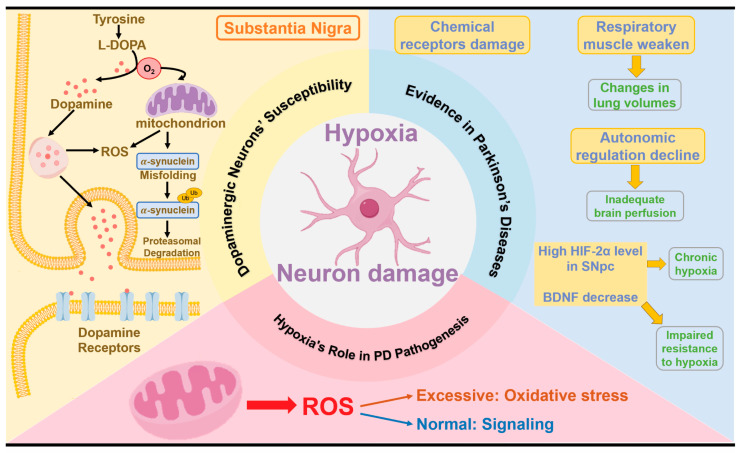
Hypoxia-induced neuronal damage and PD. Three pieces of evidence of hypoxia-induced neuronal damage and Parkinson’s disease: (1) the susceptibility of dopaminergic neurons to hypoxia, (2) clinical evidence of hypoxia in PD patients, and (3) the impact of cellular oxygen deficit on PD progression. Abbreviations: ROS: reactive oxygen species; L-DOPA: L-3,4-dihydroxyphenylalanine; Ub: ubiquitin; HIF-2α: hypoxia inducing factor-2α; SNpc: substantia nigra pars compacta; BDNF: brain-derived neurotrophic factor.

**Figure 4 ijms-25-10484-f004:**
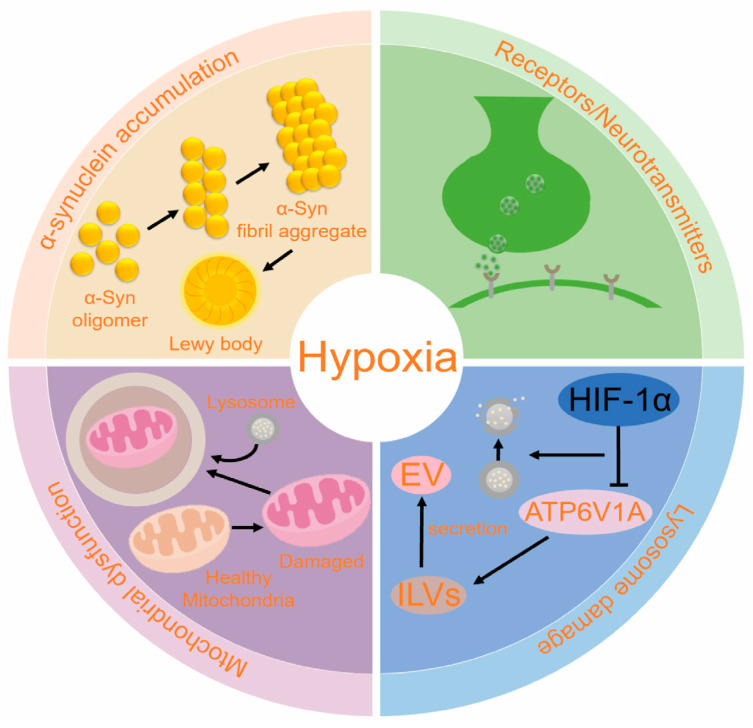
Hypoxia contributes to PD. HIF-1α influences the development of PD through multiple pathways, including the accumulation of α-syn, mitochondrial dysfunction, lysosomal damage, and alterations in receptor and neurotransmitter systems. Abbreviations: HIF-1α: hypoxia inducing factor-1α; EV: extracellular vesicle; ILVs: intracavitary vesicles; ATP6V1A: ATPase H^+^ transporting V1 subunit A; α-syn: α-synuclein.

**Figure 5 ijms-25-10484-f005:**
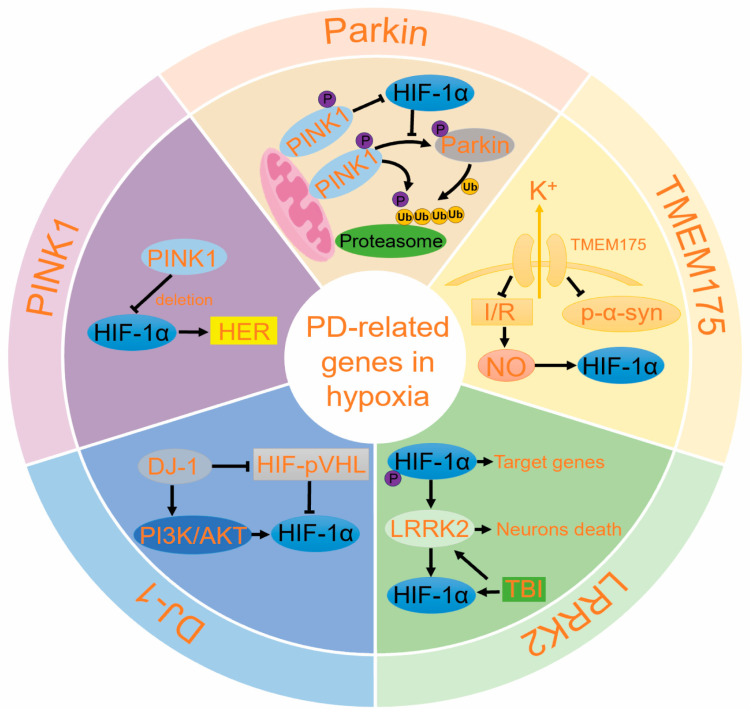
The role of five PD-related genes in hypoxia. The PRKN, PINK1, LRRK2, DJ-1, and TMEM175 genes play significant roles in the onset and progression of PD. Under certain conditions, these genes interact with HIF-1α, influencing the disease’s development. Abbreviations: PRKN: Parkin RBR E3 ubiquitin protein ligase; LRRK2: Leucine-rich repeat kinase 2; PINK1: PTEN-induced putative kinase 1; DJ-1: PARK7, Parkinson protein 7; TMEM175: Transmembrane Protein 175; HER: hypoxia response elements; HIF: hypoxia inducing factor; pVHL: von Hippel–Lindau; PI3K: phosphatidylin-ositol-3-kinase; AKT: protein kinase B, PKB; TBI: traumatic brain injury; p-α-syn: phosphorylated α-Syn; I/R: ischemia–reperfusion; NO: nitric oxide; Ub: ubiquitin; Proteasome: 26S proteasome.

## Data Availability

No new data were created or analyzed in this study. Data sharing is not applicable to this article.

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
