# Peer review of "Hypoxia Pathways in Parkinson’s Disease: From Pathogenesis to Therapeutic Targets"

_ijms, 2024, doi:10.3390/ijms251910484_

Round 1

Reviewer 1 Report

Comments and Suggestions for Authors

The manuscript (IJMS-3219983) titled “Hypoxia Pathways in Parkinson’s disease: from pathogenesis to therapeutic targets" by Gao et al. is a fascinating review that analyzed hypoxia as a potential factor in the development of Parkinson's disease.

Although the problem of the participation of hypoxia in the pathogenesis of PD is controversial, the collection of all the experimental data to date and their obvious presentation enriched with appropriate graphics has a very high cognitive value.

Considering the problem of hypoxia in the development of new, more effective PD therapies may also have practical significance in the clinic. Hence, I believe that this work deserves publication.

Reviewer 2 Report

Comments and Suggestions for Authors

Gao et al analyze the significance of hypoxia pathways in Parkinson's disease (PD). I have the following comments regarding this work:

1. The introduction lacks a description of clinical manifestation of Parkinson's disease indicating its motor and non-motor symptoms. This could possibly highlight the link between clinical and biochemical grounds - Ref.

Significance of dysautonomia in Parkinson's Disease and atypical parkinsonisms. Neurol Neurochir Pol. 2024;58(2):147-149. doi: 10.5603/pjnns.98678. Epub 2024 Mar 19. PMID: 38501557.

Diffusion tensor metrics, motor and non-motor symptoms in de novo Parkinson's disease. Neuroradiology. 2024 Aug 27. doi: 10.1007/s00234-024-03452-6. Epub ahead of print. PMID: 39190159.

2. The aspect of hypoxia could be more emphasized in the context of neurodegeneration, it would be interesting to provide a perspective on its significance in atypical parkinsonisms, where synucleinopathies and tauopathies can be mentioned.

3. The issue presented could be stressed in the context of future perspectives and possible therapies.

  • justify more extensively the choice for PD in the context of hypoxia analysis
  • provide an overview on the methodological search in the context of literature
  • provide an additional paragraph on most recent advances in the context of hypoxia in Parkinson’s Disease 
  • due to the multiple theories regarding the pathogenesis of PD e.g inflammatory, it would be interesting to provide an elaboration on the issue, this could be additionally discussed in the context of related disorders e.g. PSP and MSA

Round 2

Reviewer 2 Report

Comments and Suggestions for Authors

I do not have further comments.